# Convergent Validity of the Timed Walking Tests with Functional Ambulatory Category in Subacute Stroke

**DOI:** 10.3390/brainsci13071089

**Published:** 2023-07-18

**Authors:** Alex Martino Cinnera, Serena Marrano, Daniela De Bartolo, Marco Iosa, Alessio Bisirri, Enza Leone, Alessandro Stefani, Giacomo Koch, Irene Ciancarelli, Stefano Paolucci, Giovanni Morone

**Affiliations:** 1Santa Lucia Foundation, Scientific Institute for Research, Hospitalization and Health Care (IRCCS), 00179 Rome, Italy; a.martino@hsantalucia.it (A.M.C.); sere.marrano@gmail.com (S.M.); marco.iosa@uniroma1.it (M.I.); giacomo.koch@unife.it (G.K.); s.paolucci@hsantalucia.it (S.P.); 2Department of Human Movement Sciences, Faculty of Behavioural and Movement Sciences, Amsterdam Movement Sciences & Institute for Brain and Behaviour Amsterdam, Vrije Universiteit Amsterdam, 1081 HV Amsterdam, The Netherlands; 3Department of Psychology, Sapienza University of Rome, 00185 Rome, Italy; 4Villa Sandra Institute, Via Portuense, 798, 00148 Rome, Italy; alessiobisirri@gmail.com; 5School of Allied Health Professions, Faculty of Medicine and Health Sciences, Keele University, Staffordshire ST5 5BG, UK; e.leone@keele.ac.uk; 6Centre for Biomechanics and Rehabilitation Technologies, Staffordshire University, Stoke-on-Trent ST4 2DF, UK; 7Department of System Medicine, Faculty of Medicine and Surgery, University of Rome Tor Vergata, 00133 Rome, Italy; stefani@uniroma2.it; 8Department of Neuroscience and Rehabilitation, University of Ferrara and Center for Translational Neurophysiology of Speech and Communication (CTNSC), Italian Institute of Technology (IIT), 44121 Ferrara, Italy; 9Department of Life, Health and Environmental Sciences, University of L’Aquila, 67100 L’Aquila, Italy; irene.ciancarelli@univaq.it (I.C.); giovanni.morone@univaq.it (G.M.); 10San Raffaele Institute of Sulmona, Viale dell’Agricoltura, 67039 Sulmona, Italy

**Keywords:** stroke, gait, walking speed, outcome measures, gait disorders, neurologic, correlation of data

## Abstract

Determining the walking ability of post-stroke patients is crucial for the design of rehabilitation programs and the correct functional information to give to patients and their caregivers at their return home after a neurorehabilitation program. We aimed to assess the convergent validity of three different walking tests: the Functional Ambulation Category (FAC) test, the 10-m walking test (10MeWT) and the 6-minute walking test (6MWT). Eighty walking participants with stroke (34 F, age 64.54 ± 13.02 years) were classified according to the FAC score. Gait speed evaluation was performed with 10MeWT and 6MWT. The cut-off values for FAC and walking tests were calculated using a receiver-operating characteristic (ROC) curve. Area under the curve (AUC) and Youden’s index were used to find the cut-off value. Statistical differences were found in all FAC subgroups with respect to walking speed on short and long distances, and in the Rivermead Mobility Index and Barthel Index. Mid-level precision (AUC > 0.7; *p* < 0.05) was detected in the walking speed with respect to FAC score (III vs. IV and IV vs. V). The confusion matrix and the accuracy analysis showed that the most sensitive test was the 10MeWT, with cut-off values of 0.59 m/s and 1.02 m/s. Walking speed cut-offs of 0.59 and 1.02 m/s were assessed with the 10MeWT and can be used in FAC classification in patients with subacute stroke between the subgroups able to walk with supervision and independently on uniform and non-uniform surfaces. Moreover, the overlapping walking speed registered with the two tests, the 10MeWT showed a better accuracy to drive FAC classification.

## 1. Introduction

Stroke is a cerebrovascular disorder characterized by the sudden onset of clinical signs and symptoms [1] and represents the second leading cause of death and a major contributor to disability worldwide [2]. Intracerebral or subarachnoid hemorrhage represent 1/3 of strokes, whereas cerebral ischemia represents the remaining 2/3. Among ischemic strokes there are cardioembolic strokes and atherothrombotic strokes, both requiring hospitalization in the acute phase and associated with a high mortality risk [3]. Furthermore, the short-term prognosis of these two types of ischemic stroke is poor compared to that of other ischemic strokes. About half of stroke survivors experience severe and significant long-term daily life disability, such as difficulties with eating, bathing, and working, as well as participating in social activities [4,5,6]. Motor impairment of the lower limb is common after stroke and represents the most disabling aspect affecting the autonomy of these patients [7,8]. Indeed, walking dysfunction occurs in more than 80% of stroke survivors [9], resulting in long-term gait impairment which impacts the stroke survivor’s quality of life [9,10]. To minimize this, recovery of functional mobility and walking function is a priority of the rehabilitation programs offered to people after stroke [11]. Given the clinical importance of walking, a standardized assessment is required [12]. To assess the patient’s abilities, clinicians use disability assessment scales [13] and residual motor function scales [14]. However, the clinical motor assessment scales available today for post-stroke patients fail to assess their actual walking ability, and the walking parameters they do assess may not be truly representative of the patient’s disability status [15]. To assess walking ability, many different scales, tests and clinical instruments have been proposed. The gold standard for the investigation of gait impairment is the stereophotogrammetric gait analysis combined with electromyography; the use of force platforms and wearable inertial devices have also recently become frequently used [15,16,17]. Despite the possible disadvantages of using clinical scales or timed tests, they are still measures of first choice among healthcare professionals. Therefore, while most clinical assessments are still based on these scales, it is important that they are adequate to quantify the post-stroke patient’s deficit in a simple but accurate way. Among the clinical scales used, the Functional Ambulation Category (FAC) is one of the most common and simple tools to be used for people with locomotor deficit. The FAC scale distinguishes six levels of walking ability based on the amount of physical support required, with scores ranging from 0 (non-functional ambulation) to 5 (independent ambulation on any surface) [18,19]. This easily allows categorization of a patient’s level of ambulation and is a familiar scale for many clinicians of different training. Among the most commonly used timed walking tests, there are the 10-m walking test (10MeWT) [20] and the 6-minute walking test (6MWT) [21]. In our previous study, congruency was found between walking speed (measured during 10-m walking test) and walking capacity (measured by 6-minute walking test), and together, these may be useful to assess the safety of patients with respect to the risk of repeated falls in the community, at the point of hospital discharge from a subacute rehabilitation unit [22].

The Functional Ambulation Category and walking tests based on comfortable walking speed are each valid measures of functional mobility in adults with stroke. Characterized by a limited number of items and ease of use, FAC results are strongly correlated to walking speed and endurance, with excellent test-retest reliability and intra-rater reliability among peer assessors [23]. Despite this, there are some aspects of this scale that deserve a deeper analysis, especially for patients not requiring physical assistance. In fact, the differences may be unclear among scores 3 (no need of physical assistance but requiring supervision of a guarding person for safety and verbal cueing), 4 (independent walking on level surface, requiring supervision for negotiating stairs and non-level surfaces) and 5 (independent walking also on stairs and non-level surfaces). Furthermore, the use of these scores could be prone to subjective interpretation and can be influenced by the level of caution needed by the therapist and the self-confidence of the patient. Currently, a deep analysis of the convergent validity between FAC-score and walking tests in subacute stroke is lacking [23]. Convergent validity refers to how closely a score (in our case, the FAC score) is related to the results of other tests that measure the same or similar construct (in our case walking speed measured by 10MeWT and 6MWT). The aim of this study was to evaluate the convergent validity between the FAC test and the 10MeWT and 6MWT, by assessing the stratification of the three levels of the FAC scores related to walking ability. We tested the correlation between the FAC score and gait speed of the two walking tests, finding a cut-off value helpful to pilot the clinical evaluation of the FAC score in patients able to walk independently (with or without supervision). This may meet the need of identifying objective parameters to classify similar patients with the same FAC-score independently by their walking confidence or by the physiotherapist’s caution. Specific cut-off points determined by instrumented estimation of walking speed to optimize the convergent validity of the FAC test with walking tests, could also provide objective criteria to improve the reliability of clinical assessments of patients. Furthermore, clinicians may wish to combine the results of these timed tests (10MeWT and 6MWT with the score of the FAC) to obtain a more objective, combined, multidimensional evaluation tool.

## 2. Materials and Methods

All individuals admitted with stroke to our hospital between September 2018 and December 2020 were invited to participate in the study. Inclusion criteria included (1) first- ever stroke in the sub-acute stage (<180 days from stroke), confirmed with brain imaging test (Computerized Tomography (CT) or Magnetic Resonance Image (MRI)); (2) age between 18 to 85 years and (3) ability to walk. Exclusion criteria were: (1) concomitant lower peripheral motor neuron lesion or orthopedic disease in the lower limbs and (2) presence of moderate to severe cognitive impairment (assessed through the neuropsychological evaluation with the Mini-Mental State Examination < 24 [24]). A cross-sectional evaluation of walking ability was performed using the following tests: FAC, 10MeWT and 6MWT. Demographic characteristics (i.e., age) and clinical characterization via Barthel Index (BI) and Rivermead Mobility Index (RMI) were collected. All scales and tests were administered by an expert clinician with more than ten years of experience in the field of neurorehabilitation. All participants were stratified into three functional groups with respect to the FAC score obtained (FAC III, IV or V), in which FAC III includes ambulators, dependent on supervision, FAC IV includes independent ambulators on level surface only and FAC V includes independent ambulators. After stratification, each patient underwent the 10MeWT and 6MWT with a pause of ten minutes between two evaluations, with random first test assignment [25].

The 10MeWT was conducted at a comfortable gait speed, following a verbal start command when the patient is instructed to walk at a self-selected speed, using whatever walking aids might be needed, such as a walker or cane. Timing was recorded between 2 and 12 m on a total linear distance of 14 m [20,26,27]. The velocity was calculated as distance divided by time. For the 6MWT, participants walked unassisted in a hallway for six minutes. Instructions were scripted as “Walk as fast as comfortable for a period of six minutes. You are allowed to rest as much as you need, but time will not be stopped”. Distance was measured at the end of 6 minutes [21,28,29]. Following the participant along the entire duration of the test, the assessor does not use other words of encouragement to influence the patients’ walking speed. The protocol was approved by the local independent ethics committee, and all participants provided written informed consent prior to enrolment.

### Statistical Analysis

The statistical analysis was performed using SPSS software (version 25, IBM, Armonk, CA, USA). All continuous data are summarized here as mean ± standard deviation (SD) and dichotomous data are reported as percentage. The normal distribution of data for each parameter was verified using the Kolmogorov–Smirnov test applying the Lilliefors correction for continuous data distribution [30] and considering as significant level the highest critical value *p* = 0.20. All normally distributed data (*p* > 0.20) were analyzed using a one-way analysis of variance (one-way ANOVA), while non-normally distributed data were analyzed using the Kruskal–Wallis test. The Kruskal–Wallis test for non-parametric statistics was used to compare the three subgroups (FAC III; IV; V) in terms of their clinical scores. All results with *p* value *<* 0.05 were considered significant and investigated with a post-hoc analysis. For the post-hoc analysis has been choosing the Bonferroni correction for multiple comparisons, (with a level of significance of *p* < 0.0166) and the Dwass–Steel–Critchlow–Fligner (DSCF) test for parametric and non-parametric data respectively. Pearson and Spearman correlation analyses were performed on walking speed tests and FAC scores for parametric and non-parametric data respectively. The value of correlation coefficient was categorized as: negligible, from 0 to 0.3; low, from 0.3 to 0.5; moderate, from 0.5 to 0.7; high, from 0.7 to 0.9 and very high from 0.9 to 1, according to Cohen (1988) [31]. Following the correlations, a receiver-operating characteristic (ROC) analysis was conducted on gait speed and FAC. We reported the area under the curve (AUC) and its *p* value. We classified the AUC as follows: 0.5 < AUC ≤ 0.7 indicates lower precision, 0.7< AUC ≤ 0.9 indicates mid-level precision, 0.9 < AUC < 1 indicates high precision, and AUC = 1 indicates complete test [32]. The cut-off values for walking features with respect to the FAC score were calculated using a receiver-operating characteristic (ROC). The Youden’s index (J) was used to identify the best discriminatory cut-off value in the curve’ coordinates via the following formula:J = Sensitivity + Specificity − 1(1)

After the cut-off definition, all data were analyzed with a contingency table. The accuracy (ACC) was calculated to reflect predictiveness. The Matthew’s correlation coefficient (MCC) was used to obtain both (negative and positive) prediction values. ACC and MCC were respectively calculated as follows:ACC = (Tp + Tn)/(Tp + Tn + Fp + Fn) (2)
MMC = [(TpxTn) − (Fp*Fn)]/√[(Tp + Fp)(Tp + Fn)(Tn + Fp)(Tn + Fn)],(3)
where Tp are the true positive results; Tn are the true negative results; Fp are the false positive results and Fn are the false negative results.

## 3. Results

Eighty sub-acute stroke patients were recruited (age: 64.54 ± 13.02 years; 42.5% women; 80.06 ± 35.97 days from stroke [ranged from 19 to 180 days]; 50 with ischemic stroke [70% PACS; 14% TACS; 10% POCS; 6% LACS]) (complete demographic characteristics of the sample and the subgroups are available in Table 1). The Kolmogorov–Smirnov test showed a non-normal distribution for all variables except for age [0.07; *df* = 79; *p* > 0.20]. Following the distribution analysis result, the Kruskal–Wallis test was used to investigate the differences between the FAC subgroups (III vs. IV; IV vs. V; and III vs. V). Statistical analysis showed significant differences existed between each of the administered clinical tests across these three subgroups. These differences underline that the FAC score is related to other clinical scores as well as walking speed. In contrast, no statistical differences between FAC subgroups were observed in the demographic and baseline clinical characteristics, age, gender, lesion side, and stroke type (hemorrhagic or ischemic), which confirms overlapping in these characteristics among the three subgroups of study. Consequently, homogeneity across FAC subgroups indicates that these variables did not have an impact on FAC grouping. In contrast, the post-hoc analysis via DSCF test revealed a significant difference between each subgroup of FAC with respect to Barthel Index and RMI, but not in terms of stroke onset. With the latter, only the FAC III subgroup statistically differed with respect to the other subgroups of FAC IV and V.

### 3.1. Correlation Analysis

Non-parametric Spearman (*rs*) correlation analysis was used to investigate the nature of the relationship between two variables. The correlation analysis showed a significant moderately positive correlation between the FAC score and the walking speed assessed with the administration of the 10MeWT and the 6MWT (*rs* = 0.67, *p* < 0.001; *rs* = 0.59, *p* < 0.001 respectively). Furthermore, a high correlation was shown between the walking speed and the two tests (*rs* = 0.927, *p* < 0.001) (Figure 1). No other statistically significant correlations were found between walking speed and FAC score with respect to the demographic characteristics, confirming the trend observed in the statistical comparison of averages.

### 3.2. Receiving Operating Curve and Cut-Off Value

ROC analysis showed a mid-level precision (AUC > 0.70) with statistical significance (*p* < 0.05) in all comparisons of the FAC scores with respect to the walking speed (Figure 2A–D and Table 2).

Despite the mid-level precision in the capacity of both walking speed tests to detect a change in the FAC level, the accuracy was smaller for the 6MWT than for the 10MeWT. Furthermore, the accuracy of the 6MWT is greater between FAC III and FAC IV, than it is between FAC IV and FAC V (0.58); however, the accuracy for the 6MWT is lower than the 10MeWT in both comparisons. This finding was confirmed both in positive direction, and in bidirectional prediction (positive-negative) as indicated by a lowest value of the Matthew’s correlation coefficient (MCC) (0.40). All results and the cut-off values identified via Youden’s index are reported in Table 2.

### 3.3. Ischemic vs. Hemorrhagic

Investigating clinical data, comparing the hemorrhagic and clinical subgroups, we did not find any statistical difference. However, the walking speed analysis shows a different trend between the two subgroups and has been investigated with Wilcoxon rank-sum test for unpaired samples. Post-hoc analysis revealed a statistical difference in walking speed in both tests (10MeWT and 6MWT), exclusively in the subgroup of patients in the FAC V subgroup. Specifically, among patients able to walk independently, those who suffered a hemorrhagic stroke had a higher speed for both short and long distances compared to patients with ischemic stroke (all data are available in Table 3).

## 4. Discussion

The present study aimed to investigate the convergent validity of two walking speed tests with FAC evaluation. The two-timed walking tests were highly correlated with each other, and moderately correlated with FAC scores. Despite these highly significant correlations, and the fact that the three FAC levels divides patients into three groups, each group walking with significantly different average walking speeds, there was an overlap among FAC scores in terms of walking speeds. The objective of our analysis was to identify the best cut-off values, by optimizing the clustering of subjects by their FAC scores. In fact, when the participants were grouped based on their FAC score, we found very similar values of walking speed between the two tests, except for subjects classified in the FAC V group. In fact, despite substantially overlapping AUC values, the accuracy and the positive-negative prediction revealed that the walking speed over a short distance (ten meters) is more sensitive to detect bidirectional classification in the FAC IV and V subgroup compared to the walking speed over a longer distance. These subjects, being the most independent, can maintain their speed for six minutes as confirmed by mean speed; therefore, the incongruence in the cut-off value revealed an unsatisfactory bidirectional accuracy. Moreover, these two subgroups (FAC IV and V) can walk independently in a relatively short time after their stroke, especially compared to the FAC III subgroup. This information can be useful because it unmasks information that is not apparent from only comparing walking speeds over long distances. One possible explanation for the reduced accuracy level of the long-distance walking speed is that the results depend on the subject functional characteristics. The 6MWT was originally used to evaluate cardiopulmonary capacity [33], and later used in neurorehabilitation to assess walking capacity. As demonstrated in a previous work, subjects affected by subacute stroke may manage long-distance walking using various methods [34]. In fact, walking after stroke is demanding in terms of energy expenditure, even for less-affected subjects [34,35]. Some patients decrease velocity, especially in the second half of the test, to manage the latero-lateral oscillations of their trunk, while others maintain a stable velocity, accepting an increment of the trunk oscillation [36,37]. However, the latter motor behavior, while it leads to a higher walking capacity, is more correlated with the risk of falls. In a previous study, this incongruence in long-distance walking and the higher risk of fall was noted [22]. Notwithstanding, gait speed is fundamental in the prognosis of community ambulation outcomes among inpatients discharged from stroke rehabilitation care [38]. Our results suggest that these are not the only parameters that impact walking ability when assessed over long distances.

In summary, in the 10MeWT we found useful cut-off values (0.59 m/s and 1.02 m/s) with a good bidirectional predictive value with respect to all FAC subgroups considered. This positive convergent validity can support the gait evaluation of subacute patients, especially in the differentiation between the higher level of walking performance. In line with this observation, both FAC score and speed on the short distance are predictors of fall risk [39,40].

However, it should be reported that the patients able to walk everywhere independently (allocated in the FAC V subgroup) were highly heterogeneous and there was a statistical difference in walking speed between the etiopathogenesis of stroke (ischemic vs. hemorrhagic). Patients with hemorrhagic stroke with the maximum score of FAC showed a high average speed in both tests compared to subjects with ischemic stroke in the same FAC subgroup, notwithstanding an overlap in other clinical and demographics variables. This observation is reported previously in the literature, where hemorrhagic stroke showed a greater improvement in gait skills, and specifically in walking speed [41].

In the literature, walking measurements (functional walk distances and self-paced speed) were correlated with balance function, stroke specifics and global impairment score [42]. Our results are important because they add knowledge to the use of the short distance walking velocity tests in a specific pathological population, and because they are the most simple, inexpensive, and commonly used tests. A large body of literature agrees with the fact that a reduced walking speed is generally correlated to a major risk for disability, cognitive impairment, institutionalization, falls and mortality [43].

In summary, walking ability is fundamental for the patient’s social participation when returning home after hospital and its assessment with simple scales and tests is important. Our data support the use of the FAC test and the 10MeWT especially in high level walking patients in the subacute phase of stroke, before social reintegration to potentially reduce their risk of falls.

### 4.1. Limitations

Our results are not generalizable to all stroke populations; in fact, the subacute phase of stroke differs from the chronic phase and is characterized as those without a stable functional status, with basic motor intentions and actual motor actions that are slightly misaligned. This complex relationship calls into question the locomotor body schema and its potential and necessary neuroplasticity modification during the recovery after stroke [44]. Present conclusions are based on collapsed data of hemorrhagic and ischemic stroke, albeit reproducing the distribution of pathogenesis of stroke [45] as in the general population. However, the recovery patterns are somewhat different (i.e., in walking speed). In the present study, the limited number of patients included in the subgroups (ischemic and hemorrhagic) did not allow for separated ROC analysis based on FAC scores and walking speed. Moreover, all scales and tests were assessed by only one clinician; thus, inter-rater reliability of the FAC test could not be calculated. However, there are reports in the literature that the FAC scale has good inter-rater reliability (κ = 0.72) [46]. This could have mitigated possible errors resulting from lack of agreement between multiple raters. Despite this, to confirm the current deductions, future studies should consider the above-mentioned limitations.

### 4.2. Future Perspective

Instrumental evaluation of walking speed would be useful to investigate the convergent validity with FAC test in future research. Specifically, the use of cutting-edge technologies would support investigators in a more accurate quantification of cut-off values which can be used in clinical practice [47,48,49]. Moreover, the correlation between instrumented walking speed cut-offs and other clinical scales (i.e., Fugl–Meyer Assessment scale, Berg Balance scale, National Institute of Health Stroke scale) will provide additional information about the involvement of motor, sensory, and joint functions and gait balance skills. Additionally, inter-rater reliability could be provided to evaluate the potential for bias in administration. Finally, stratification on a larger sample could reveal different characteristics about gait performance and more precise cut-off values. In particular, the differentiation between ischemic and haemorrhagic stroke and between lacunar and non-lacunar ischemic stroke can provide new insight about the correlation of pathophysiology and clinical gait features in these populations. This differentiation is important given the different clinical features reported for lacunar and non-lacunar stroke [50,51].

## 5. Conclusions

The evaluation of walking velocity is crucial in the routine assessment of functional status of patients following subacute stroke, and in designing personalized neurorehabilitation programs to improve post-discharge outcomes. From this investigation we found a good convergent validity between 10MeWT and FAC test scores, with a clear cut-off in terms of walking speed (0.59 and 1.02 m/s). The assessment on this short distance can be used to drive the attribution of the highest level of FAC score. In contrast, the convergent validity was lower between FAC score and the walking speed assessed over a long distance than in the 6MWT for FAC IV and V subgroups. This information suggests that it is necessary to carefully investigate patients with high functional levels over long distances, especially in the hemorrhagic stroke subgroup, who show a higher average speed compared to ischemic patients. Moreover, FAC IV and V subgroups start to walk independently about at the same time following stroke, making it more difficult to use the onset variable to drive their allocation. The assessment of walking autonomy using the FAC test is fundamental to describe the independency of patient; however, when there are some doubts about the FAC score, clinicians could evaluate the timed assessment of walking speed on a short linear distance (such as by the 10MeWT) to differentiate the gait level of the patient.

## Figures and Tables

**Figure 1 brainsci-13-01089-f001:**
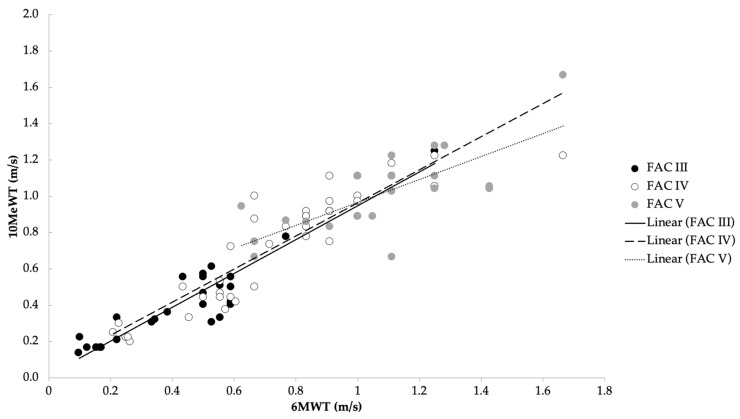
Correlation between average speed in the 10MeWT and 6MWT with respect to the three functional levels of FAC (*rs* = 0.97).

**Figure 2 brainsci-13-01089-f002:**
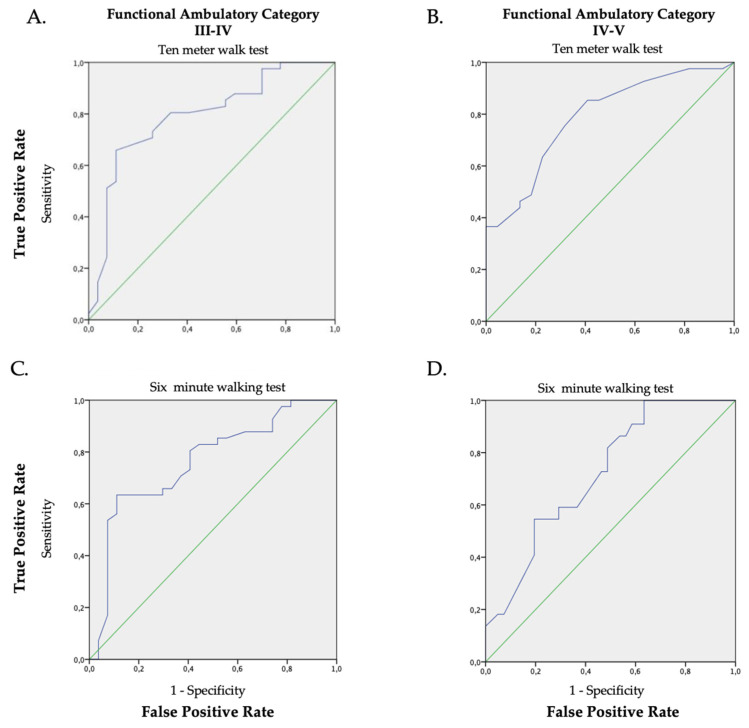
ROC graphs of the walking speed (average), the 10MeWT (Panel (**A**,**B**); AUC 0.79 *p* < 0.001 and AUC 0.79 *p* < 0.001) and the 6MWT (Panel (**C**,**D**); AUC 0.76 *p* < 0.001; AUC 0.72 *p* = 0.004) in relation to the FAC classification (Panel (**A**,**C**): FAC III vs. FAC IV; Panel (**B**,**D**): FAC IV vs. FAC V). The green line is the random classificatory, while the violet one is the prediction of the trade-offs between the true and false positive based on our data.

**Table 1 brainsci-13-01089-t001:** Demographic characteristics.

	Total Sample	FAC III	FAC IV	FAC V
Sample (n)	80	24	35	21
Male/Female (n)	46/34	11/13	21/14	14/7
Age (years) ^a^	64.54 ± 13.02	66.13 ± 13.36	65.03 ±12.85	61.95 ± 13.2
Isch./hemor. (n)	50/30	18/6	19/16	13/8
Side: Right/Left (n) ^b^	43/37	14/10	15/20	14/7
Onset ^a^	80.06 ± 35.97	108.05 ± 45.05 **	71.32 ± 23.96 *	62.55 ± 19.92 *
Barthel Index ^a^	86.7 ± 18.19	70.29 ± 19.82 **	91.51 ± 14.43 **	97.95 ± 3.73 **
RMI ^a^	10.57 ± 3.7	6.74 ± 3.16 **	11.57 ± 2.58 **	13.19 ± 1.57 **
FAC ^a^	3.96 ± 0.75	3	4	5
10MeWT (s) ^a^	/	35.46 ± 27.85 **	15.94 ± 10.43 **	9.87 ± 2.82 **
10MeWT speed (m/s) ^a^	/	0.45 ± 0.28 **	0.79 ± 0.31 **	1.08 ± 0.27 **
6MWT (m)^a^	/	160.71 ± 101.81 **	284.35 ± 110.57 **	367.24 ± 83.41 **
6MWT speed (m/s) ^a^	/	0.45 ± 0.28 **	0.79 ± 0.31 **	1.02 ± 0.23 **

Abbreviations: n, number; SD, standard deviation; FAC, Functional Ambulation Category; RMI, Rivermead Mobility Index. * Statistically significant with only one subgroup (IV vs. III and V vs. III specifically); ** Statistically significant respect to the other two subgroups; ^a^ result expressed as mean ± standard deviation; ^b^ affected hemisphere.

**Table 2 brainsci-13-01089-t002:** Results of convergent validity analysis of walking speed tests with respect to functional walking levels of FAC.

	Cut-Off	AUC	*p*-Value	Sensitivity	Specificity	ACC	MCC
FAC III vs. FAC IV							
10MeWT (m/s)	0.59	0.79	<0.001	0.89	0.34	0.75	0.54
6MWT (m/s)	0.66	0.76	<0.001	0.63	0.11	0.74	0.52
FAC IV vs. FAC V							
10MeWT (s); ws (m/s)	9.76; 1.02	0.79	<0.001	0.85	0.41	0.76	0.46
6MWT (m); ws (m/s)	216; 0.6	0.72	0.004	1	0.63	0.58	0.40

Abbreviations: 10MeWT, 10-m walking test; 6MWT, six minutes walking test; FAC, Functional Ambulation Category; ACC, Accuracy; MCC, Matthew’s correlation coefficient; m/s, meters per second.

**Table 3 brainsci-13-01089-t003:** Clinical difference between ischemic and hemorrhagic subgroups. We report statistical significant values as marked with an asterisk (*) for *p* ≤ 0.05.

	Ischemic (50)	Hemorrhagic (30)	*p*-Value
Male/Female (n)	31/19	15/15	
Age (years) ^a^	65.98 ± 12.58	62.13 ± 13.60	0.20
Onset (days) ^a^	81.12 ± 40.28	78.42 ± 28.80	0.76
Barthel Index ^a^	84.5 ± 20.61	90.37 ± 12.72	0.16
RMI ^a^	10.24 ± 3.9	11.1 ± 2.78	0.30
FAC ^a^	3.9 ± 0.78	4.06 ± 0.69	0.34
10MeWT ws ^a^	0.72 ± 0.38	0.87 ± 0.36	0.07
6MWT ws ^a^	0.69 ± 0.35	0.85 ± 0.34	0.06
Post-hoc analysis			
FAC III 10MeWT ws ^a^	0.43 ± 0.31	0.51 ± 0.17	0.55
FAC III 6MWT ws ^a^	0.42 ± 0.30	0.51 ± 0.19	0.51
FAC IV 10MeWT ws ^a^	0.78 ± 0.33	0.81 ± 0.30	0.77
FAC IV 6MWT ws ^a^	0.77 ± 0.31	0.80 ± 0.30	0.78
FAC V 10MeWT ws ^a^	1.00 ± 0.25	1.24 ± 0.21	0.04 *
FAC V 6MWT ws ^a^	0.93 ± 0.18	1.17 ± 0.20	0.01 *

Abbreviations: 10MeWT, 10-m walking test; 6MWT, six minutes walking test; RMI, Rivermead Mobility Index; FAC, Functional Ambulation Category; ws, walking speed; ^a^ result expressed as mean ± standard deviation.

## Data Availability

The data that support the findings of this study are available on request to the corresponding author.

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
