# Peer review of "Convergent Validity of the Timed Walking Tests with Functional Ambulatory Category in Subacute Stroke"

_brainsci, 2023, doi:10.3390/brainsci13071089_

Round 1

Reviewer 1 Report

The authors investigate in eighty walking participants with first-ever stroke in subacute stage the convergent validity of two walking speed tests with Functional Ambulation Category (FAC) evaluation. The authors found that two-timed walking tests were highly significantly correlated with each other, and moderate significantly correlated also with FAC scores. Moreover, the overlapping walking speed registered with the two tests, the 10-meter walking test (10MeWT)  showed a better accuracy to drive FAC classification. The use of FAC and the 10MeWT especially in high level walking patients in the subacute phase of stroke before social reintegration and potentially reduce their risk of falls. The authors concluded that the assessment of walking autonomy by FAC is fundamental to describe the independency of patient, and when there are some doubts about the score, the clinicians could evaluate the  timed assessment of walking speed on a short linear distance (such as by the 10MeWT) to differentiate the level of autonomy of the patient. The study is potentially  interesting, but can be improved if the following considerations are ad-dressed:  

 1.It would be helpful to mention in the Introduction that cardioembolic stroke and atherothrombotic stroke are  the subtypes of ischemic infarct with the highest in-hospital mortality and the short-term prognosis of patients with cardioembolic stroke and atherothrombotic stroke is poor compared with other ischemic stroke subtypes.

2. It would be interesting to know the different ischemic stroke subtypes in the study population.

3. The authors should indicate that an essential line of research in the future would be precisely the assessment of the evidence relating convergent validity of the timed walking tests with functional ambulatory category in lacunar versus non-lacunar ischemic stroke. This recommendation is because the pathophysiology, prognosis, and clinical features of lacunar strokes are different from other acute cerebrovascular diseases (Int J Mol Sci 2022; 23, 1497). We recommend including and commenting on this reference.

4. Please check references #7, and #14.

Author Response

The authors would like to thank each one of the reviewers for the time spent on our manuscript. Their qualified suggestions helped us to improve it in its revised and resubmitted version. In the following, our point by point responses are reported in blue font with new parts of text between apices in italics style.

Please see the attached file for authors' response.

Reviewer 2 Report

The paper was overall well-written and presented. Here are some suggestions:

1. The significance, especially clinically, of establishing the convergent validity between the walking tests and the Functional Ambulation Category was not sufficiently explained.

2. It is unclear how the 10-meter walking test was conducted. The order of the two tests was randomized, but unclear whether the participants were allowed to rest after one test and how much rest time was unspecified.

3. There is a lack of assessment of motor function. The article could be strengthened with motor functioning, balance, sensation, and joint functioning using the Fugl-Meyer Assessment. 

Author Response

(The authors gave the same response as above.)

Reviewer 3 Report

The article “Convergent Validity of the Timed Walking Tests with Functional Ambulatory Category in Subacute Stroke” is written very well. I practically do not find anything that could be changed, added ... However, in table 2, the 6MWT parameter is better to transfer its dimension (m / s) to another line. Just like it was done for the 10MeWT parameter. It will be easier to perceive the meaning. And vice versa, for table 3 in the section - Post-hoc analysis, the first two parameters - index "a" should be moved to the same line. So, as it is done for the next 4 parameters.

Author Response

(The authors gave the same response as above.)
